# An Integrative Review of Recovery Services to Improve the Lives of Adults Living with Severe Mental Illness

**DOI:** 10.3390/ijerph18168873

**Published:** 2021-08-23

**Authors:** Eric Badu, Anthony Paul O’Brien, Rebecca Mitchell

**Affiliations:** 1School of Nursing and Midwifery, Faculty of Health and Medicine, The University of Newcastle, Callaghan, NSW 2308, Australia; 2Faculty of Health, Southern Cross University, East Lismore, NSW 2480, Australia; tony.obrien@scu.edu.au; 3Faculty of Business and Economics, Macquarie University, Macquarie Park, NSW 2109, Australia; rebecca.mitchell@mq.edu.au

**Keywords:** recovery, rehabilitation, systematic reviews, mental health, nursing

## Abstract

There is an increasing call for recovery-oriented services but few reviews have been undertaken regarding such interventions. This review aims to synthesize evidence on recovery services to improve the lives of adults living with severe mental illness. An integrative review methodology was used. We searched published literature from seven databases: Medline, EMBASE, PsycINFO, CINAHL, Google Scholar, Web of Science, and Scopus. Mixed-methods synthesis was used to analyse the data. Out of 40 included papers, 62.5% (25/40) used quantitative data, 32.5% used qualitative and 5% (2/40) used mixed methods. The participants in the included papers were mostly adults with schizophrenia and schizoaffective disorder. This review identified three recovery-oriented services—integrated recovery services, individual placement services and recovery narrative photovoice and art making. The recovery-oriented services are effective in areas such as medication and treatment adherence, improving functionality, symptoms reduction, physical health and social behaviour, self-efficacy, economic empowerment, social inclusion and household integration. We conclude that mental health professionals are encouraged to implement the identified recovery services to improve the recovery goals of consumers.

## 1. Introduction

Recovery in mental health is defined as the process of restoration of functioning and well-being, improvements in quality of life of an individual diagnosed with longer-term mental health problems or emotional difficulties, using a whole-system or integrated approach [1,2,3,4]. The concept of recovery has been defined according to two main categories—clinical recovery (e.g., service-based, or objective definition) and personal recovery (subjective) [1,2,3,4]. The two forms of recovery—clinical and personal—are relevant in physical and mental conditions. The personal perspective considers recovery as a journey or process, whilst the clinical perspectives consider it as an outcome [2,5]. Personal recovery is a continuous process involving personal growth and development, improvement of symptoms, regaining control, and establishing a personally fulfilling and meaningful life, and incorporates restoration to a healthy state [1,3,5]. The concept also encapsulates a growing sense of agency and autonomy [5]. Some studies have defined personal recovery as pursuing one’s personal hopes, aspirations, personal abilities, and in the achievement of a personally acceptable quality of life [1,6,7]. Furthermore, some evidence has suggested that personal recovery should be given attention to provide holistic mental health services [4,8]. In particular, the individual’s values and preferences for specific treatments or other forms of support should be central [4]. Conversely, clinical recovery, which is based on a medical perspective, includes remission of symptoms or a return to normal functioning and is less holistic [1,3,9].

A broad range of recovery services are being promoted worldwide in hospitals and community settings. Such services are operationalized as interventions that aim to provide person-centred mental health services, thus a multidimensional process of transformation involving positive transitions. The primary aim of recovery services is to help individuals to recover and develop some social and intellectual skills needed to live, learn and work in the community [2,5,6,7]. Recovery services help to improve psychological, social and cognitive functioning, social inclusion, and positive thinking of people with mental illness [1,2,3,4,10,11]. Despite this, past evidence has shown that several multi-level systemic and individual consumer factors continue to impede recovery services [2,3,4,5]. The systemic factors are mostly associated with the strengths of psychiatric facilities (patient loads, time constraints in clinical encounters), the capacity of mental health professionals (ancillary professionals and training support), governance or leadership of mental health services (management resistance to change and comprehensive multidisciplinary and inter-service collaboration), affordability, and inclusivity of the mental health services [2,3,4,5,11]. Additionally, individual factors include the ability to ensure early access to services, and normative life pursuits such as education, employment, sexuality, friendship, spirituality, and religious participation [2,5].

Several empirical studies regarding recovery services are being conducted recently to promote the personal recovery journey of consumers. The evidence has largely addressed services used to foster integrated recovery services. Only a few of such studies have attempted to systematically review the literature regarding recovery services for adults with severe mental illness (SMI). A preliminary search we conducted showed that though some existing reviews have been conducted regarding recovery, they are limited to the general mental health population and some specific services, such as individual placement services [10,12,13]. Critically, no review study has been undertaken to aggregate a synthesis of both qualitative and quantitative studies regarding recovery services that promote the personal recovery journey of adults living with severe mental illness. Adults with serious mental illness (SMI) may suffer from a long-term condition that requires a special integrated or whole-system approach to treatment. This review, therefore, aimed to: (1) identify existing evidence on recovery services; and to (2) synthesize evidence on the outcome of recovery services for adults living with severe mental illness.

The review findings are significant for several reasons. The evidence is expected to inform policy decision making on the well-being of adults with serious mental illness. The evidence is also considered to be valuable to policy makers and mental health professionals to strengthen mental health services. Finally, the evidence can guide researchers and clinicians in terms of future research and further inform the training of mental health professionals and students.

## 2. Methods

### 2.1. Methodology

An integrative review is an approach that allows simultaneous inclusion of diverse methodologies (e.g., qualitative and quantitative data) and varied perspectives to fully understand the phenomenon of concern [14,15]. This review study aims to use diverse data sources to develop a holistic understanding of recovery services for adults with severe mental illness. This review method contributes greatly to evidence-based practice for mental health nursing. This review employed a five-stage process—problem identification (developing and defining research question and study aim); searching literature (incorporating a comprehensive search strategy); evaluation of data (assessing for methodological quality); analysis of extracted data (data reduction, display, comparison and conclusions) and presentation (mixed-methods synthesis implications for practice, policy and research) [14].

### 2.2. Inclusion Criteria

This review included studies that address recovery services for adults (e.g., 18 years and above) living with severe mental illness (e.g., in-patients, out-patients, community-based residential services, home-based services). This review defined individuals with serious mental illness as those with a mental, behavioural, or emotional disorder resulting in serious functional impairment, which substantially interferes with or limits one or more major life activities [16]. Adults with serious mental illness were individuals with schizophrenia, bipolar disorder, mania, or psychosis that have been diagnosed by a health professional and self-reported or by proxy [17]. Studies that were included targeted services such as disability support and recovery services. This study also included papers that address the effect of recovery services on the lives of people living with severe mental illness.

This review included papers of all methods and designs. Papers included used mixed methods as well as quantitative and qualitative methods. The quantitative methods included quantitative randomized controlled trials, quantitative non-randomized designs (analytical cross-sectional) and quantitative descriptive studies. Additionally, the qualitative papers used ethnography and participatory methodology, grounded theory, phenomenology, and narrative. This review only considered studies published in the English language from January 2008 to January 2020—a period of increased research into recovery services and interventions for adults with mental illness.

### 2.3. Exclusion Criteria

Papers that were excluded are based purely on general health services or clinical effectiveness of a particular intervention with no connection to recovery services and mental health rehabilitation. Additionally, papers were excluded if they address recovery services for children and adolescents, workplace mental health issues, recovery services in stroke patients or traumatic injury. Other general exclusion criteria were systematic reviews, conference abstracts, clinical case reviews, book chapters, papers that present opinions, editorials and commentaries.

### 2.4. Search Strategy and Selection Procedure

We searched seven electronic databases—EMBASE, CINAHL (EBSCO), Web of Science, Scopus, PsycINFO, Medline and Google Scholar. The search was conducted according to the Joanna Briggs Institute (JBI) recommended guidelines for conducting systematic reviews [18]. A three-stage search strategy was utilised to search for information (see Table 1). An initial limited search was conducted in EMBASE and MEDLINE (Table 1). The initial search was not restricted by limiters—field, language, timespan and type of publication. We analysed the text words contained in the title and abstract and the index terms from the initial search results [18]. A second search using all identified keywords and index terms was then conducted across all remaining databases. These searches were restricted to title, abstract and keywords due to a plethora of references obtained through the initial search. Finally, the reference lists of all identified articles were hand searched for additional studies [18].

The selection of eligible articles adhered to the Preferred Reporting Items for Systematic Reviews and Meta-Analyses (PRISMA) [19] (Figure 1). Firstly, three authors independently screened the titles of articles that were retrieved and then approved those meeting the selection criteria. The authors reviewed all titles and abstracts and agreed on those needing full-text screening. The first author conducted the initial screening of titles and abstracts. The second and third authors conducted the second screening of titles and abstracts of all the identified papers. The authors conducted full-text screening according to the inclusion and exclusion criteria.

### 2.5. Data Management and Extraction

Endnote X8 (software) was used to manage the search results, screening, reviewing articles, as well as removing duplicate references. Three reviewers independently managed the data extraction process [18]. The authors developed a data extraction form to handle all aspects of data extraction (Appendix A). The data extraction form was developed according to the Cochrane and the Joanna Briggs Institute (JBI) manuals [18] for systematic reviews as well as consultation with experts in methodologies and the subject area [18]. The authors extracted results of the included papers in numerical, tabular and textual format [18]. The first author conducted the data extraction whilst the second and third authors conducted the second review of the extracted data. The data extraction focused on study details (citation, year of publication, author, contact details of lead author, and funder/sponsoring organization), publication source, methodological characteristics, study population, subject area (e.g., recovery service model, recovery concept, recovery intervention, period of project implementation, phases or components of recovery intervention, outcome or impact of intervention), as well as additional information and recommendations and other potential references to follow up.

### 2.6. Assessment of Methodological Quality

The authors developed a critical appraisal checklist using the Mixed-Methods Appraisal Tool (MMAT) [20] and the Joanna Briggs Institute (JBI) [21] critical appraisal tool. The critical appraisal checklist was used by the authors (the first, second and third authors) to assess the methodological quality of the included papers. The critical appraisal tool was sub-divided into sections such as study details, methodology (e.g., categorized as qualitative, quantitative randomized controlled trials, quantitative non-randomized, including cohort study, case–control study, analytical cross-sectional, quantitative descriptive, and mixed methods) and overall quality score (Appendix A). The methodological quality score was rated as low quality if the overall score was below 25%, medium quality if 50% and high quality 70% and above. The scores were computed by summing the number of ‘Yes’ counts in each sub-section of the methodological criteria. The total score was then expressed as a percentage [20].

## 3. Data Synthesis

The extracted data were analysed using mixed-methods synthesis [14,18]. Mixed-methods synthesis seeks to develop an aggregated synthesis of qualitative and quantitative data [18]. The process involves familiarization with the data, generating initial codes, searching for themes, reviewing themes, defining and naming themes and producing a thematic chart [22,23]. The authors coded the quantitative and qualitative data together. Data display matrices were developed to document all of the coded data from each extracted data [14]. Alphabets and colours were assigned to each of the coded ideas. The resulting codes from quantitative and qualitative data were used to generate descriptive themes [18]. The descriptive themes were categorized into global and organizing themes. The themes have been discussed with the concepts and theoretical constructs that explain recovery services in mental health. The background information of included papers was analysed using STATA version 15.

## 4. Results

### Description of Retrieved Papers

This review retrieved 788 papers from all databases. Of these, 94 duplicates were removed. The titles and abstracts of 694 non-duplicate articles were screened for eligibility, after which 266 were excluded. A total of 428 full-text articles were assessed for eligibility (376 were excluded). This review extracted data from 52 full-text articles that met the eligibility. Of these, 2 articles were identified through hand searching the reference list. Overall, 40 papers were included in the final synthesis (Figure 1). Out of the 40 papers, 38 met the criteria for high methodological quality assessment, whilst only two papers had medium quality.

## 5. Characteristics of Included Articles

More than half of the papers (23/40; 57.5%) were interventional studies. Of these, more than one-third (12/29; 43.47%) used randomized controlled trials. More than half of the included papers (25/40; 62.5%) used quantitative methods, 32.5% used qualitative methods and 5% (2/40) used mixed methods. The participants in the included papers were mostly adults with schizophrenia and schizoaffective disorder (Table 2). Approximately 57.5% (23/40) of the included papers used descriptive and inferential statistics, 17.5% (7/40) used thematic analysis, and 5% (2/40) each used descriptive statistics alone, grounded theory and triangulation (thematic, descriptive and inferential statistics). In addition, 27.5% (11/40) of the included papers were studies conducted in the USA, 10% (4/40) were studies conducted across six European centres (UK, The Netherlands, Germany, Italy, Bulgaria and Switzerland), 10% (4/40) were conducted in Sweden, 7.5% (3/40) targeted Hong Kong and 5% (2/40) each focused on Canada, China, South Africa and the Netherlands (see Table 2).

## 6. Context for Implementing Recovery Services

This review identified five environments or contexts where recovery services are implemented (Table 3). Environments where recovery services were implemented were communities, residential facilities and services via psychiatric hospital and primary health care settings [25,26,30,31,44,57] (Table 3). Four papers suggested that recovery services can be offered through home-based care [33,34,36,45] and day centre structure programs [28,37,56].

## 7. Mechanisms for Implementing Recovery Services

### 7.1. An Integrated Recovery Service

In this review, an integrated recovery service model is described as any services that seek to promote and support restoration, ‘remediation and reconnection’. The concept employs both an overarching, inherently collaborative and integrated approach to mental health services. Most of the review papers (16/40) described integrated recovery models as services used to achieve the personal recovery of adults with severe mental illness (Table 3). Most of the papers suggested that integrated recovery services can be delivered through illness management [47,48,53,58,61,62,63], mindfulness-based interventions [46], task-sharing or task-shifting approaches [25,26], home visits [24,40,44], active leisure or recreational activities [34,56] and music therapy [30] (see Table 4). The reviewed papers highlighted that integrated recovery services generally aim at developing independent living skills, improving quality of life, community mobilisation [48], reducing inpatient and crisis services, adhering to treatment and setting meaningful goals towards recovery [24,53,58].

Integrated recovery services can be offered through training sessions (e.g., hours, days and weekly for several months) [25,26,30,44,46,53]. For instance, past study regarding integrated recovery was enhanced with mindfulness group therapy sessions which were run for 60 min throughout 26 weeks [46] (see Table 4). Conversely, music-creation therapy used as a recovery service was run for 32 weekly sessions, with a duration of 90 min for each session [30]. Generally, the activities of integrated recovery service focused on cognitive behaviour therapy techniques, psychoeducation, relapse prevention, social and coping skills training (meals, guidance in activities of daily living, role-playing, hobby groups) [24,47,53], adherence support, family therapy, crisis management, household contribution and understanding medication [24,25,44]. More specifically, Tjornstrand, Bejerholm [56] recommended that active leisure implemented as recovery services can include activities—playing sports; the opportunity to play games, eat, and socialize; embarking on excursions; relaxation (see Table 4).

Two studies concluded that conventional rehabilitation services can incorporate additional innovative interventions aimed to achieve recovery for consumers [30,47]. For instance, Luk [47] recommended the inclusion of spiritual intervention (a form of hymn singing, Bible reading, personal sharing and intercessions) in conventional rehabilitation services. Similarly, Chang, Chen [30] recommended the use of music-creation therapy as a recovery service for adults with SMI. These recovery services are delivered by different mental health professionals, including clinical psychologists, community health workers, psychiatrists, occupational therapists, social workers and counsellors [25,26,40,44,47]. Some studies further suggested that non-specialists are sometimes trained to deliver recovery services through task-sharing or task-shifting approaches. Some of the non-specialist professionals are auxiliary social workers [25,47].

### 7.2. Vocational Rehabilitation (Individual Placement Services)

Eighteen of the included papers recommended several vocational rehabilitation programs used to improving the lives of consumers (see Table 3). These are Individual Placement and Support (IPS) [28,29,33,34,43,57,59,60], supported employment enterprises, sheltered employment [27,34,51,57], conventional vocational rehabilitation [59] and social firms [51,54].

Most of the included papers described the process of implementing Individual Placement and Support. The papers suggested that the IPS is implemented through phases such as initial vocational assessments (e.g., to identify clients’ strengths and work skills), job searching (e.g., searching job sites, applying for work online and accompany clients to interviews), individual job development (e.g., intensive supervision), work performance monitoring, support for employers and continuing post-employment support for clients [27,28,29,33,59]. In addition, some papers recommended that Individual Placement and Support workers receive training and regular supervision to provide effective services [29,60] (see Table 4). IPS employment can take several forms, including services (e.g., cleaning, gardening, catering, clerical and administrative work) [56,57], training clients for the labour market, agricultural production and creative projects (e.g., painting, drawing, sculpture, ceramics and textiles, assembly lines, carpentry shops, computer repair centres, bicycle repair shops woodworking and furniture making) [39,54].

### 7.3. Recovery Narrative Photovoice, Art Making and Exhibition

Five of the included papers recommended photovoice, art making and exhibition as interventions used to construct recovery [32,35,41,49,50]. Photovoice, art making and exhibition are used to achieve recovery, empowerment, community integration [32,49,50] and share difficult experiences non-verbally [35]. The intervention aims to explore, document and share ideas about recovery. It involves the construction of text with photographs through the exhibition and large group discussion [32,49,50]. More importantly, the intervention helps to avoid the stigma that is associated with the conventional process of delivering mental health services [41]. For instance, Clements [32] suggested that readers or audience of photovoice interventions become part of the construction of the meaning of recovery. The intervention allows people with serious mental illness to communicate their needs and ideas to the public, as well as to policy makers.

Photovoice, art and exhibition interventions are delivered through weekly class sessions and community outings [41,49,50] (Table 3). The content of the class sessions focus on writing exercises, psychoeducational handouts, and activities on how to construct empowering narratives of recovery and stigmatization [49,50] (see Table 4). In addition, Ketch, Rubin [41] suggested that the class sessions begin with sharing previous experiences or knowledge about art making. The final outcomes of the photovoice, art and exhibition interventions are documented through the final recovery photo gallery, text pieces, art shows, public exhibition, creative arts (e.g., painting, ceramics, silk screening, and mosaics) and displays of art prints [32,35,41,50].

## 8. Outcome of Recovery Services

### 8.1. Psychiatric Medication and Treatment

Ten of the included papers highlighted that recovery services have helped to improve the clinical outcomes of adults living with SMI [24,25,33,36,42,44,48,53,62,63]. The services specifically increased access to psychiatric medication and antipsychotic medication adherence, decreased relapse, improved knowledge and illness management, and decreased clinical contact [24,25,33,36,42,44,53]. For instance, Lee, Liem [44] reported that recovery services have improved most outcome parameters such as bed days, readmission episodes and missed psychiatric appointments. Conversely, Malinovsky, Lehrer [48] suggested that the number of days spent in the hospital decreased by approximately 40% after initiation of recovery transformation. Furthermore, some studies suggested that the effects of IPS interventions on the time patients spent in competitive employment have had a significant effect on the clinical status, particularly a reduced need for psychiatric inpatient care [33,42]. For instance, Kilian, Lauber [42] indicated that consumers who received an IPS intervention spent fewer days in the hospital.

### 8.2. Improve Functionality

Fourteen of the included papers concluded that recovery services have improved the functioning of adults living with severe mental illness (Table 3). Recovery services improved social and psychological functioning as well as motor and process abilities of adults living with severe mental illness. More specifically, Asher, Hanlon [24] reported in a study that a CBR intervention improved functioning in adults with schizophrenia (baseline median: WHODAS = 57.5 and IQR 36.7, 65.1; end line median: WHODAS = 18.4 and IQR 2.4, 46.2). Similarly, Zhou, Zhou [63] showed in a study that the intervention (e.g., rehabilitation training programs such as day treatment, medication monitoring, biweekly rehabilitation training) group had a significant improvement in social and psychological functioning.

### 8.3. Reduce Symptoms

Most of the included papers suggested that recovery services have had a significant improvement in anxiety, psychosocial and mental illness symptoms of adults with severe mental illness [25,30,31,36,42,44,45,46,53,57,63]. For instance, Chang, Chen [30] reported that anxiety symptoms in an experimental group (music-creation program) improved 6.22 points more than in the control group (*p* < 0.001). Similarly, the mean symptoms (Positive and Negative Syndrome Scale PANSS) in a clubhouse group (e.g., open occupation or employment) decreased from 64.5 to 42.7 compared with the control group—from 51.7 to 57.6 (*p* = 0.01) [57]. More importantly, Lopez-Navarro, Del Canto [46] concluded that incorporating mindfulness interventions into integrated rehabilitation has the potential to reduce negative symptoms.

### 8.4. Improvement in Physical Health and Social Behaviour

Six of the included papers reported that recovery services have improved the physical health and social behaviour of adults with severe mental illness [24,26,36,45,46,48]. In particular, recovery services have shown improvements in physical health, well-being, adaptation, appearance [24,36,45,48], and quality of life (psychological health) [46] as well as reductions in risk-taking behaviour [26]. For instance, Lopez-Navarro, Del Canto [46] recommended that incorporating mindfulness-based interventions into recovery services can improve psychological health-related quality of life. More specifically, the study indicated that mindfulness interventions account for 38% of the variance in health-related psychological quality of life [46].

### 8.5. Self-Efficacy

Twenty-one of the included papers described the impact of recovery services on the self-efficacy or self-reliance of adults with severe mental illness (see Table 3). The papers highlighted that recovery services improved self-esteem or self-confidence (e.g., fostered feelings of self-worth) [26,33,35,47,49,50,51,57,61], hope [24,25,30,32], improvements in thoughts, emotions and better understanding of mental illness [25,26,33]. More so, recovery services improved self-care or practical skills of daily life (e.g., bathing, washing clothes, as well as undertaking chores related to food preparation and household cleaning) [26,32] as well as personal empowerment [32,49,50,51].

Three papers recommend that recovery narrative photovoice and art-making services have positively impacted adults with severe mental illness. Such recovery services have improved the lives of consumers in areas through impacts such as on their sense of identity and independence as well as their ability to tolerate uncertainty, take ownership of choices, learn from the past, and maintain vigilance for relapse [35,49,50]. In particular, art-making interventions provide opportunities for self-exploration and the development of new skills [35]. More importantly, art making and exhibition help people living with severe mental illness clearly express their feelings and communicate emotions and thoughts, which can be difficult to express using words. In addition, recovery narrative photovoice interventions also have a positive impact on outcomes such as autonomy, readiness for change, inspiration, idealism, transformation of self, acceptance of support, awareness of progress, hope, determination, passion, perseverance, introspection, strength, and sense of connectedness [49,50].

Moreover, four papers recommended that vocational rehabilitation interventions impacted positively on sense of identity (e.g., sense of competence through participation in work), reducing boredom/loneliness, staying anchored in reality and creating strong internal motivation for change [39,43,51,52]. The feelings and expressions of clients help them to develop a sense of self-determination and personal recovery [52,54].

### 8.6. Economic Empowerment

Nineteen of the included papers reported that recovery services, for instance, vocational participation, have helped to improve the economic empowerment of adults with severe mental illness (see Table 3). More importantly, vocational rehabilitation programs provide livelihood and income-generating activities [24,26,27,43,51].

Most papers reported that adults living with severe mental illness who participated in vocational interventions gained competitive employment [27,31,36,56,60], returned to open employment [28] and received vocational benefits [38,43,56,60]. In particular, adults living with severe mental illness participating in vocational interventions are more likely to receive job-seeking assistance (e.g., searching for jobs, filling application forms and practice for interviews) [43,47,60].

Most of the studies suggested that vocational interventions such as IPS are more effective than the conventional vocational services, particularly in every vocational outcome. IPS clients are more effective at working competitively, returning to open employment (e.g., working for at least one day), and achieving a longer duration of employment (e.g., working for many hours and longer job tenure) and securing greater wages [28,29,31,42,43,59]. Catty, Lissouba [29] reported that IPS clients were two-fold (214 days) more likely to work for a longer duration than vocational service clients (108 days). Conversely, 57% of IPS clients (a sample of 58 consumers with schizophrenia) worked competitively, compared with 29% of conventional vocational clients. Similarly, 70% of IPS participants obtained any paid work, compared with 36% of conventional vocational clients [59]. More importantly, vocational rehabilitation interventions have helped consumers to gain financial literacy skills (e.g., managing finances) [24,25,26,33,51], achieve financial independence and stability [43] and recover [24,31].

### 8.7. Social Inclusion (Community Integration)

Twenty-seven of the included papers reported that recovery services have increased social inclusion and community acceptance or integration of adults with severe mental illness (see Table 3). Specifically, recovery services achieve increasing social and community participation (e.g., participating in social activities such as churches, coffee ceremonies weddings and funerals), reduce discrimination [24,56], reduce social isolation, create supportive social environments [25], increase social contact or social interaction [26,39,41,43,44,45,55,56] and increase socialization (e.g., being around and having breaks and playing games) [41,55,56].

Some studies reported that recovery services such as IPS recovery narrative photovoice and art making help adults with severe mental illness to achieve or re-establish valued roles in the community [33,35,49]. Whitley, Harris [61] recommended that consumers with severe mental illness use the community as a place of safety and surrogate family as well as for socialization and individual growth. In addition, vocational participation rehabilitation services increase the social contact between adults with SMI and their supervisors and customers or clients, which subsequently reduces the feeling of social isolation [39,43,51,54,55].

Furthermore, some studies suggested that social environment recovery services, (e.g., clubhouse used as a community) can create an atmosphere of acceptance and inclusion and subsequently support each member’s personal recovery journey [52,54]. For instance, adults with severe mental illness in residential programs (clubhouse) showed greater participation in recreational events and informal socialization with peers [62], more social relationships, greater quality of life [57,63] and feeling valued, and greater inclusion and belonging to a group [38]. Further, De Heer-Wunderink, Visser [34] reported that supported independent living programs seemed to positively influence the level of social inclusion for consumers living with severe mental illness in terms of being active and socializing with others.

### 8.8. Household Integration

Three of the included papers reported that recovery services have achieved integration of adults with severe mental illness into their families [24,25,36]. Such recovery services have increased greater participation in household tasks and family stability and care [24,25]. For instance, Asher, Hanlon [24] reported that a recovery-oriented rehabilitation service has helped adults with severe mental illness return home to address the basic needs of shelter and food. The services have also equipped family caregivers to develop resilience to accommodate their relatives, including informing them of plans in advance, communicating calmly and avoiding stressors. Consequently, the services have helped to reduce caregiver burden as well as treating adults with severe mental illness with dignity and effective caregiving (e.g., providing food, shelter and shelter) [24].

### 8.9. Social Support

Seven of the included papers highlighted several support services used to implement recovery services for adults with SMI [31,33,34,36,39,50]. Some papers highlighted that support services originate from sources such as relatives, friends or peers [34,39] as well as supervisor support and community peers [33,39]. In particular, De Heer-Wunderink, Visser [34] suggested that more than 85% of clients in recovery-oriented rehabilitation service reported having received support from a partner, their family, or friends. Conversely, Harpaz-Rotem, Rosenheck [36] reported that clients receiving a residential treatment had significantly higher social support on average (*p* < 0.001) after baseline. Supervisors (staff) from recovery services provide practical and emotional support to adults with severe mental illness [39,61]. Whitley, Harris [61] reported that most adults with severe mental illness considered supervisors or staff to be equally important members of their surrogate family.

## 9. Discussion

This review was conducted to synthesize evidence on recovery services used to improve the lives of adults living with severe mental illness. The review findings are discussed according to two emerging themes: (i) mechanisms for implementing recovery services and (ii) outcome of recovery services.

### 9.1. Mechanisms for Implementing Recovery Services

Recovery services are interventions that aim to provide person-centred mental health services, through a whole-system or holistic approach towards the recovery journey of consumers [5]. The review findings identified several recovery services including integrated recovery services, vocational rehabilitation and recovery narrative photovoice and art-making exhibitions. Integrated recovery services, for instance, are offered through illness management, mindfulness interventions, and task-shifting approaches (e.g., participatory-based training) as well as home visiting, active leisure and music therapy services. Integrated recovery services are mostly incorporated into conventional services and aim to achieve holistic mental health services. The review findings encourage service providers to integrate mindfulness practices, active leisure, music therapy and spiritual healing practices as part of integrated recovery service models [30,46,56]. The inclusion of such components not only ameliorates the symptoms but is also useful in achieving a sense of agency and autonomy, taking personal responsibility and getting on with life [6]. Conversely, integrated recovery services encourage the use of task shifting and home visiting to promote the recovery journey [24,25,26,40,44]. Home visiting and task shifting can help consumers take up a central role in managing illness (e.g., regaining control) and personal growth as well as establishing a fulfilling and meaningful life [1]. Task shifting and home visiting can help consumers and their families to set personal goals towards their recovery journey. Our review findings encourage service providers to implement integrated recovery service models to improve the lives of consumers.

The evidence suggests that vocational rehabilitation services are also increasingly employed to promote the recovery process of consumers This service is mostly offered through IPS, supported employment enterprises and social firms compared with conventional vocational services (e.g., sheltered employment) [28,29]. Vocational rehabilitation services that are used to support the recovery process of consumers are consistent with previous literature [10,12,13]. In particular, IPS interventions have been proven to improve the recovery journey. IPS interventions identified in the current findings are mostly associated with agricultural production, creative projects, services and clients joining the labour market [28,29,43,57]. The interventions are usually implemented through initial vocational assessment, job searching, individual job development, monitoring work performance, support for employers and continuing post-employment support for clients. The review findings encourage service providers to implement vocational rehabilitation services (e.g., IPS interventions) that are contextually respected by local service providers and communities. More specifically, researchers are encouraged to use interventional studies to measure the effectiveness of different vocational rehabilitation services that aim to promote the recovery process of consumers.

Recovery narrative photovoice, art-making and exhibition interventions have recently been employed as recovery services to support the recovery process [32,35,41,49,50]. Photovoice, art making and exhibition services are implemented through text construction and photographs. This service is also presented through exhibition and large group discussion. This service is particularly employed to achieve recovery, empowerment, community integration, express difficult experiences in non-verbal forms as well as avoid the stigma associated with conventional mental health services. The review findings encourage researchers to use interventional studies to explore the effectiveness of recovery narrative photovoice and art-making exhibition services towards the recovery journey, particularly in multi-cultural settings, where there is increased stigmatization towards mental illness. Such interventional studies could help consumers to develop culturally sensitive recovery goals.

### 9.2. Outcome of Recovery Services

Recovery services necessarily focus on enhancing consumer’s capacities for living with, managing, and pursuing a life in the presence of disability, as well as removing barriers around their environment [7]. Consistent with earlier studies, current recovery services are useful in enhancing psychiatric medication and treatment (clinical outcomes) of consumers [3,7]. More specifically, such services increase access to psychiatric medication, antipsychotic medication adherence, decrease relapse, improve knowledge and decrease clinical contact [24,25,33,36,42,44,53]. These services are not only about helping consumers to learn how to live a fuller and more satisfying life but also contribute to the reduction in the symptom itself. The review findings encourage service providers to promote holistic care that considers the individual’s subjective appraisal of his or her functioning and satisfaction with life [8].

In addition, psychiatric rehabilitation has recently moved beyond the mere control of symptoms and prevention of relapse to incorporate a functional recovery and enhancement of the quality of life of the consumer [1]. The ability of service providers to improve the quality of life of consumers could help to achieve a personal recovery process that is consumer-centred. The current review findings demonstrated that recovery services have supported the physical health and social behaviour of consumers. More specifically, such recovery services improve physical health, well-being, adaptation, appearance, and quality of life and reduce risk-taking behaviour [24,26,36,45,46,48]. Improvement in the physical and social behaviour of consumers could also help them to progress in developing self-efficacy, self-confidence, and gaining hope, improvements in self-care or practical skills. The interventions have specifically enhanced the recovery process through readiness for change, autonomy, inspiration, idealism, sense of connectedness and transformation of self.

Recovery services have also improved the economic empowerment of adults living with severe mental illness. The services provide livelihood and income-generating avenues that can facilitate access to competitive employment, returning to open employment and vocational benefits. It is apparent that the participation in income-generating activities improves the financial literacy skills, financial independence and financial stability of adults living with severe mental illness. The improvement in finances through income-generating activities forms a major component of the recovery journey or process. The findings confirm previous literature which suggest that recovery services could empower adults with severe mental illness, through normative life participation such as education, social and political activities [1,2]. The major strength of recovery services mostly relies on its ability to safeguard empowerment in the consumers through everyday living skills, accommodation, social networks, employment and education endeavours [2]. Again, the review findings demonstrated that recovery services aid social inclusion and community acceptance of adults living with severe mental illness. Consistent with previous literature, recovery-oriented rehabilitation promotes the inclusion of adults with severe mental illness through increasing community participation, social contact or social interaction and socialization as well as creating supportive social environments [3]. Such services support consumers to reconnect and re-establish a place in the community, and to explore opportunities that could help them live an independent life. Consequently, the services reduce social isolation, discrimination and stigmatization among consumers [2]. In addition, consumers are also integrated greatly into their individual families and so take an active role in family activities and also maintain family stability.

## 10. Limitations

This review has several limitations that need consideration. The limitations of the integrative review are largely pertinent to the search words, language limitations, and period of included papers. This review was limited to papers published in the English Language from January 2008 to January 2020. In particular, limiting studies to only English Language articles published between January 2008 to January 2020 could miss relevant non-English Language articles as well as those published before 2008. The variation in search terms and keywords regarding recovery services may miss some relevant articles. Additionally, each included study uses different tools for assessing the clinical and personal recovery of the included consumers. This does not allow distinguishing which intervention is most useful. In addition, participants (adults with different mental illnesses) may have different prognoses in different interventions. Given that we did not perform a meta-analysis, there is no objective measure of the effect size of included papers to determine the outcome of recovery. Despite these limitations, this study has some strengths. For example, the combination of clearly articulated search methods, consultation with a research librarian, and reviewing articles with multiple experts as well as the quality assessment tool used to measure the methodological quality helped to address the various limitations.

## 11. Conclusions

The review findings showed that several studies have been undertaken regarding recovery services that can improve the lives of adults living with severe mental illness. Most recovery services are implemented in developed western countries, particularly in the USA and Europe, with relatively few studies piloted in developing countries (for example Africa). The review findings demonstrated that most of the papers used quantitative data, with few studies employing both qualitative and quantitative data to achieve complementarity or convergence. The evidence showed that most recovery services are delivered through community-based settings. Additionally, studies on recovery services largely address issues on integrated recovery service and vocational rehabilitation, with few studies addressing recovery narrative photovoice and art-making exhibition services. Furthermore, recovery services are reported to be relevance in areas such as medication and treatment adherence, functioning, symptoms, physical health and social behaviour, self-efficacy, economic empowerment, social inclusion or community integration, household integration and access to social support services.

## 12. Implications for Mental Health Policy and Practice

Based on the findings, we recommend that awareness and advocacy for recovery services should be prioritised in national and international policy initiatives. Specifically, consumer associations, self-help groups and family caregivers could be empowered to take the leading advocacy role in recovery services. Additionally, such recovery services should be prioritised by clinicians and allied health professionals to support the recovery goals of consumers. Further research on recovery services should be prioritized in clinical practice and directed towards interventional studies, which can provide sustainable and workable solutions in the recovery journey and outcomes. Given that recovery research mostly employs quantitative and qualitative methods, we recommend that future recovery research should attempt to use mixed methods to achieve complementarity and congruence in both methods.

## Figures and Tables

**Figure 1 ijerph-18-08873-f001:**
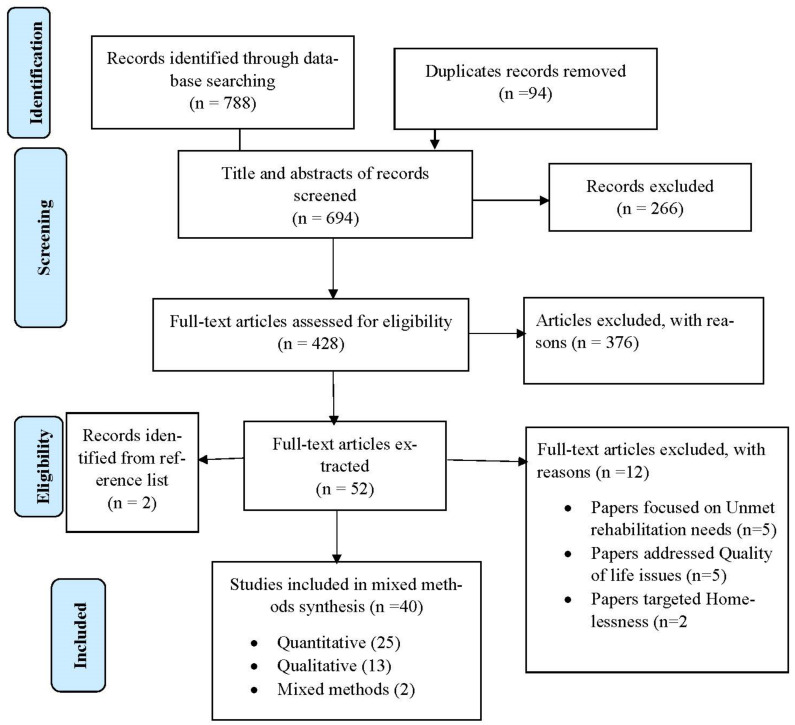
Flow chart of included papers.

**Table 1 ijerph-18-08873-t001:** Search strategy and selection procedure.

Stages	Search Terms and Keywords
Stage 1 (initial search in MEDLINE and EMBASE)	(Disabilit* or “psychosocial disability” or Adult or “mental disorders” or “mental illness” or “mental condition”).mp. AND (“Social Support” or “Individual Support” or “disability support” or “social inclusion” or integration or “community acceptance” or participation).mp. AND (“nursing homes” or “residential facilities” or “residential care” or “rehabilitation centers” or “community rehabilitation” or “residential program” or “residential care” or institutions or residential treatment).mp. AND (“service model” or “service typology” or rehabilitation or rehabilitat * or “disabled persons” or vocational or “psychosocial support” or or “psychosocial deprivation”).mp. AND (recovery or effectiv * or “patient reported” or “outcome measures” or “treatment outcome” or “patient outcome” or assessment or functioning or “quality of life” or coping or “patient-centered care”).mp
Stage 2 (search across CINAHL, Web of Science, Scopus, and PsycINFO)	(Disabilit * or “psychosocial disability” or Adult or “ental disorders” or “mental illness” or “mental condition”).mp. AND (“social support” or “individual support” or “disability support” or “social inclusion” or integration or “community acceptance” or participation).mp. AND (“nursing homes” or “residential facilities” or “residential care” or “rehabilitation centers” or “community rehabilitation” or “residential program” or “residential care” or institutions or residential treatment).mp. AND (“service model” or “service typology” or rehabilitation or rehabilitat * or “disabled persons” or vocational or “psychosocial support” or or “psychosocial deprivation”).mp. AND (recovery or effectiv * or “patient reported” or “outcome measures” or “treatment outcome” or “patient outcome” or assessment or functioning or “quality of life” or coping or “patient-centered care”).mp
Stage 3	Hand searching the reference list

**Table 2 ijerph-18-08873-t002:** Characteristics of included papers.

Papers	Objectives	Setting	Age/Gender	Participants	Study Design/Methods	Data Collection Instrument	Analysis	Q^+^	Summary of Article
Asher, Hanlon [24]	The acceptability and feasibility of CBR in practice as well as how CBR may improve functioning among people with schizophrenia	Ethiopia	Mean age = 39.5; males and females	Schizophrenia	Quasi-experimental design/Mixed methods	In-depth interviews (IDIs) Discrimination and Stigma Scale-12 (DISC-12) Alcohol Use Disorders Identification Test (AUDIT) Patient Health Questionnaire-9(PHQ-9) Involvement Evaluation Questionnaire (IEQ) World Health Organization Disability Assessment Schedule (WHODAS) 2.0 Clinical Global Impression	Thematic analysis and descriptive statistics	H	CBR program has the capacity to improve functioning of people with schizophrenia
Brooke-Sumner, Lund [25]	To develop a community-based psychosocial rehabilitation program for service users with schizophrenia	South Africa	Maximum age = 45; females	Schizophrenia	Quasi-experimental design/Qualitative	• In-depth interviews (IDIs)	Thematic analysis	H	The program improved the lives of service users with schizophrenia—self-esteem, social support, illness knowledge, self-care, and contribution to their households
Brooke-Sumner, Selohilwe [26]	Investigated a non-specialist-delivered program for psychosocial rehabilitation for service users with schizophrenia in a low-resource South African setting	South Africa	Age range = 21–44; males and females	Schizophrenia	Quasi-experimental design/Mixed methods	In-depth interviews (IDIs) Brief Psychiatric Rating Scale (BPRS) Clinician-administered scale World Health Organization Disability Assessment Scale (WHODAS) Stigma of Mental Illness Inventory (ISMI)	Thematic analysis and inferential statistics	H	The program achieved reduction in ISMI assessment as well as improved psychosocial well-being of service users with schizophrenia
Browne and Waghorn [27]	to retrospectively assess the implementation of IPS practices and youth employment outcomes	New Zealand	Age range = 16–25; males and females	Affective (including comorbid anxiety), bipolar affective disorder, generalized anxiety disorder	Observational design (retrospective case study)/Quantitative	• Case review	Descriptive statistics	M	The IPS program was effective in terms of the proportion of young clients commencing competitive employment, and duration of longest job held
Burns, White [28]	The acceptability and effectiveness of IPS in Europe	Five European countries	Mean age = 38	schizophrenia or schizoaffective disorder bipolar disorder	Randomized controlled trial/Quantitative	• Questionnaire	Inferential statistics	High	Individual Placement and Support (IPS) was approximately two-fold more effective than vocational services in returning to open employment
Catty, Lissouba [29]	Determine which patients with severe mental illness do well in vocational services and which process and service factors are associated with better outcomes	Six European centres	Age range = 18–local retirement age; males and females	Schizophrenia	Randomized controlled trial/Quantitative	Global Assessment of Functioning—Symptoms (GAF–S) and Disability (GAF–D) Positive and Negative Syndrome Scale Hospital Anxiety and Depression Scale Groningen Social Disability Schedule Lancashire Quality of Life Profile—European Version Rosenberg Self-Esteem Scale Camberwell Assessment of Need—European short version	Inferential statistics	H	The IPS services were more effective than the vocational services for every vocational outcome
Chang, Chen [30]	Investigated the effect of a music-creation group program on the anxiety, self-esteem, and quality of life of patients with SMI	Taiwan	Age range 20–65; males and females	Schizophrenia or affective disorders	Quasi-experimental design/Quantitative	Demographic data Hamilton Anxiety Rating Scale (HAM-A) Rosenberg Self-Esteem Scale (RSES) World Health Organization Quality of Life-BREF (WHOQOL-BREF)	Inferential statistics	H	Participating in a structured music-creation intervention improved the psychological well-being, self-esteem, quality of life and social relationship of consumers with SMI
Chiu, Ho [31]	To test empirically the Substance Abuse and Mental Health Services Administration (SAMHSA) recovery model	Hong Kong	Mean age = 41.6; males and females	Schizophrenia spectrum disorder	Cross-sectional/Quantitative	Internalized Stigma of Mental Illness scale (ISMI) Resilience Scale (RS) Making Decision Empowerment scale (MDES) Exercise of Self-Care Agency Scale Mastery Scale (MS) Adult State Hope Scale (ASHS) Health Care Climate Questionnaire (HCCQ) Recovery Attitude Questionnaire (RAQ-7) Medical Outcome Study Social Support Survey—Chinese version (EISS-MOS-SSS-C) Schizophrenia Quality of Life Scale (SQLS) Multidimensional Scale of Perceived Social Support—Chinese version (MSPSS-C) World Health Organization Spirituality Religion and Personal Belief Scale—Hong Kong version (WHO-SRPBHK)	Inferential statistics	H	Psychosocial symptoms, respect, resilience, and empowerment were significant contributors of recovery
Clements [32]	To pilot a PAR and photovoice project to facilitate discussions about recovery based on personal and local experience	Canada			Participatory action research/Qualitative (e.g., photovoice)	Photo/text pieces, other ‘readers’ or audiences	Recovery photo gallery	H	Photovoice proved as a useful research method for the construction of local knowledge about recovery and as a vehicle for sharing that knowledge
Crain, Penhale [33]	Application of IPS in a Canadian community mental health team through the study of a competitively employed individual and support network	Canada	Mean age = 42; males	Schizophrenia	Instrumental case study/Qualitative	• In-depth interviews (IDIs)	Thematic analysis	H	The IPS program had positive outcomes through securing and maintaining job, changing perceptions, self-confidence, social skills and recovery
De Heer-Wunderink, Visser [34]	Investigated levels of social inclusion among service users of two types of psychiatric community housing programs in the Netherlands	Netherlands	Mean age = 44; males and females	Schizophrenia, anxiety or depression, personality disorder	Cross-sectional design/Quantitative	Health of the Nation Outcome Scales (HoNOS) Camberwell Assessment of Need Short Appraisal Scale (CANSAS)	Descriptive and inferential statistics	H	Supported independent living programs seemed to positively influence the level of social inclusion among service users compared with residential programs
Fenner, Ryan [35]	Analysed what consumers and staff reported at the end of the project	Samoa			Interpretive phenomenological design/Qualitative	• Focus group discussion	IPA	H	Art making positively impacts on consumers senses of identity and independence and demonstrates their talents and capacities
Harpaz-Rotem, Rosenheck [36]	Observational data comparing 1 year clinical outcomes among women who received RT services and those who did not	USA		Psychiatric/substance abuse problems	Quasi-experimental design/Quantitative	Self-report interview Psychiatric, Alcohol, and Drug Composite Scales Short Form Health Survey (SF-12) Symptom Checklist-30 (SCL) Post-Traumatic Stress Disorder (PTSD) Symptom Checklist (PCL)	Descriptive and inferential statistics	H	Placement in residential treatment was associated with significantly improved clinical outcomes in a variety of domains
Hultqvist, Markstrom [37]	Comparing users of two approaches to psychosocial rehabilitation in Sweden, community-based mental health day centres (DCs) and clubhouses	Sweden	Mean age = 48.7; males and females	Psychoses, mood and anxiety disorders, autism/neuropsychiatric disorders	Quasi-experimental design/Quantitative	Manchester Short Assessment of Quality of Life (MANSA) Self-Esteem Rosenberg Global Assessment of Functioning (GAF) MOS 36-item Short-Form Health Survey Social Interaction (ISSI) Satisfaction with Daily Occupations (SDO) Swedish version of the CSQ-8 The EPM-DC Socio-demographic and clinical factors	Descriptive and inferential statistics		The study showed that visiting clubhouses appears to be more beneficial for improved QOL in a longer-term perspective
Hultqvist, Markström [38]	Compared DC and clubhouses, concerning the users’ perceptions of unit and program characteristics, and aspects of everyday occupations in terms of engagement and satisfaction	Sweden	Mean age = 48.7; males and females	Psychoses, mood and anxiety disorders, autism/neuropsychiatric disorders	Combined cross-sectional and longitudinal comparative study/Quantitative	Evaluation of Perceived Meaning in Day Centres (EPM-DC) Productive occupations (POES-P) Satisfaction with Daily Occupations (SDO) scale	Descriptive and inferential statistics		The users of clubhouse performed better than day centre users various on social network sub-scales (feeling valued by others, feelings of inclusion and belonging to a group)
Iancu, Zweekhorst [39]	Analysed and compared experiences of recovery on prevocational services, in order to assess if users make progress towards recovery, relative to a staged recovery model	Netherlands	Mean age = 42.5; males and females	Schizophrenia and personality disorders, depressive and anxiety disorders	Interpretive phenomenological design/Qualitative	• Semi-structured interviews	Thematic analysis	H	The prevocational services provide the needed services for people with mental disorders who desire to engage in recovery (create strong internal motivation for change)
Iwasaki, Coyle [40]	The role of leisure-generated meanings (LGMs) experienced by culturally diverse individuals with mental illness in potentially helping them to better cope with stress, adjust and recover	USA	Mean age = 48; males and females	Bipolar disorder, major depression, schizophrenia, bipolar/schizophrenic, schizoaffective disorder, substance abuse, panic disorder, borderline personality	Cross-sectional/Quantitative	Colorado Symptom Index (CSI Recovery Assessment Scale (RAS Psychosocial Adjustment to Illness Scale (PAIS Leisure Meanings Gained Scale (LMGS Leisure Coping Scale (LCS Leisure Satisfaction Scale (LSS Leisure Boredom Scale (LBS Perceived Active Living Scale (PALS)	Descriptive and inferential statistics	H	Leisure can contribute to stress-coping, recovery, adjustment, and active living for individuals with mental illness
Ketch, Rubin [41]	Art appreciation for veterans with severe mental illness in a VA Psychosocial Rehabilitation and Recovery Centre	USA			Quasi-experimental design/Qualitative	• Photos and interviews	Thematic analysis	H	The program had positive effects on mood, self-esteem, socialization community participation and recovery process of veterans with SMI
Kilian, Lauber [42]	Analyses the relationships between employment hours, psychopathological symptoms and the days of inpatient treatment detected	Six European centres	Mean age = 37.8; males and females	Schizophrenia/schizoaffective disorders, bipolar disorder	Randomized controlled trial/Quantitative	Positive and Negative Symptoms Scale (PANSS) OPCRIT	Descriptive and inferential statistics	M	IPS intervention through its effect on the time spent in competitive employment leads to reduced need for psychiatric inpatient care
Koletsi, Niersman [43]	Explore clients’ experiences of the support received from their IPS or vocational service workers and the perceived impact of work on clients’ lives	Six European centres	Age range = 18–57; males and females	Schizophrenia/schizoaffective disorders, bipolar disorder	Randomized controlled trial/Qualitative	• Semi-structured interviews	Thematic analysis	H	The IPS program improved financial stability, illness, social life, increased self-esteem, integration into society, self-improvement, coping strategy and reduced loneliness
Lee, Liem [44]	Explored the effectiveness of Assertive Community Treatment (ACT) for severely ill mental patients during a period of rapid deinstitutionalization in Hong Kong	Hong Kong	Age range = 18–65; males and females	Psychotic disorders	Flanking historical control design/Quantitative	World Health Organization Quality of Life (WHOQOL-) Hong Kong Chinese Version Brief Psychiatric Rating Scale (BPRS) Clinical Data Analysis and Reporting System (CDARS)	Descriptive and inferential Statistics	H	The ACT had positive effect over and above the conventional treatment models—outcome parameters (bed days, readmission episodes, missed psychiatric appointments, BPRS and quality of life) improved
Lindstrom, Hariz [45]	To evaluate clients’ activities of daily living (ADL) ability and health factors outcomes following their participation in occupation-centred interventions in home and community settings	Sweden	Mean age = 48; males and females	Schizophrenia, schizoaffective disorder, bipolar disorder, Asperger syndrome, obsessive compulsive disorder	Quasi-experimental design/Quantitative	Goal Attainment Scaling (GAS) Assessment of Motor and Process Skills (AMPS) Assessment of Social Interaction (Swedish version BSI-II) Satisfaction with Daily Occupations (SDO) ADL taxonomy Symptom Checklist–90 (SCL-90)	Descriptive and inferential Statistics	H	The occupational therapy services integrated in to sheltered or supported housing achieved positive lifestyle, meaningful occupations and participation in society
Lopez-Navarro, Del Canto [46]	The effectiveness of group mindfulness-based intervention (MBI) in patients diagnosed with severe mental illness	Balearic Islands	Mean age = 38.44	Schizophrenia, bipolar disorder, delusional disorder	Randomized controlled trial/Quantitative	World Health Organization Quality of Life-BREF (WHOQOL-BREF) Positive and Negative Syndrome Scale (PANSS) Mindfulness Attention Awareness Scale (MAAS)	Descriptive and inferential statistics	H	Mindfulness intervention in rehabilitation has potential to enhance quality of life and reduce negative symptoms
Luk [47]	Investigate the long-term effects of a holistic care program for the rehabilitation of persons with serious mental illness	Hong Kong	Mean age = 35; males and females	Schizophrenia, manic depressive, depression	Cross-sectional/Quantitative	World Health Organization Quality of Life Measure (WHOQOL-BREF(HK) General Health Questionnaire (GHQ) Rosenberg Self-Esteem Scale (ESTEEM) Social Support Questionnaire-6 (SSQ-6) Purpose in Life Questionnaire (PIL) Hopelessness Scale (HOPE) SSQ-Satisfaction	Descriptive and inferential statistics	H	The program is effective to provide positive changes—support, encouragement, self-confidence, spiritual assistance and reflection of values
Malinovsky, Lehrer [48]	Evaluate the effectiveness of a recovery-oriented transformation carried out by a large, private, not-for-profit psychiatric rehabilitation organization serving individuals with SM	USA	Mean age = 46.42; males and females	Schizophrenia, mood disorder (unipolar/bipolar), other psychotic disorder	Longitudinal study/Quantitative	Multnomah Community Ability Scale—Revised Clinician Rated (MCAS-R) Self-Report (MCAS-SR) Competency Assessment Instrument (CAI) State Hope Scale (SHS) Client Version (WAI-C) and Therapist Version (WAI-T)	Descriptive and inferential statistics	H	Recovery-oriented services are effective to reduce hospitalizations and improve quality of life
Mizock, Russinova [49]	Describe the development and feasibility of the recovery narrative photovoice intervention	USA		Serious mental illnesses	Community-based participatory research/Quantitative	Ryff Scale of Psychological Well-Being Empowerment Scale Community Integration Measure	Descriptive statistics		The program has the potential to facilitate recovery-related outcomes, including empowerment, positive identity, and community integration
Mizock, Russinova [50]	Explore the meaning of recovery for individuals with serious mental illness	USA		Serious mental illness	Community-based participatory research/Qualitative	• Photos and archival data	Thematic analysis	H	The study identify several internal and external recovery strategies and outcomes
Panczak and Pietkiewicz [51]	Explore personal experiences of people employed in Vocational Development Centres	Poland	Age range = 28–58; males and females	Schizophrenia spectrum disorders	Interpretative phenomenological design/Qualitative	• Semi-structured interviews	Consecutive analytical	H	The program improved the economic and social well-being of people with schizophrenia—economic empowerment, empowerment, functioning and social inclusion
Raeburn, Schmied [52]	Explore how recovery practices are implemented in a psychosocial clubhouse	Australia	Mean age = 47; males and females	Schizophrenia, bipolar disorder or schizoaffective disorder	Case study/Qualitative	In-depth interviews (IDIs) Observations—Spradley’s field note domains, and the Recovery and Promotion Fidelity Scale (RPFS)	Theoretical thematic analysis	H	The psychosocial clubhouse is a community that provide opportunity to participate in a personal recovery journey
Salyers, McGuire [53]	To rigorously test Illness Management and Recovery (IMR) against an active control group in a sample that included veterans	USA	Mean age = 47.7; males and females	Schizophrenia, schizoaffective disorder	Randomized controlled trial/Quantitative	Structured Clinical Interview for DSM-IV Positive and Negative Syndrome Scale (PANSS) Quality of Life Scale (QLS) Patient Activation Measure Morisky Scale Recovery Assessment Scale (RAS) State Hope Scale	Descriptive and inferential statistics	H	Improved significantly in a number of domains related to illness management—symptoms, psychosocial functioning, self-rated illness management, and emergency department use
Svanberg, Gumley [54]	Explore the experience of recovery from mental illness in the context of two emerging social firms	Scotland	Age range = 19–64; males and females	Bipolar disorder, depression, psychosis, anxiety, addictions	Social constructionist (grounded theory)/Qualitative	• Open-ended interview questions	Thematic analysis	H	The social firms are effective to enhance self-confidence, acceptance and inclusion of people with mental illness
Swildens, van Busschbach [55]	Investigate the effect of the Boston Psychiatric Rehabilitation (PR) approach on attainment of personal rehabilitation goals, social functioning, empowerment, needs for care, and quality of life in people with severe mental illness (SMI) in the Netherlands	Netherlands	Age = 41 and over; males and females	Schizophrenia or schizoaffective disorder, bipolar disorder, depressive or anxiety disorder, personality, addiction, cognitive disorder	Randomized controlled trial/Quantitative	Self-report Social Functioning Scale Camberwell Assessment of Need Short Appraisal Schedule WHOQOL-BREF Personal Empowerment Scale BPRS—Extended version GAF—symptoms and disabilities Client Socio-Demographic and Service Receipt Inventory—European version PR Beliefs, Goals and Practices scale Working Alliance Inventory	Descriptive and inferential statistics	H	Psychiatric rehabilitation has a significant impact on goal attainment, societal participation and social contacts
Tjornstrand, Bejerholm [56]	Gaining knowledge regarding the occupations performed in day centres, in terms of the participants’ descriptions of what they were doing	Sweden	Mean age = 45.3; males and females	Schizophrenia, other psychoses	Interpretative phenomenological design/Qualitative	Time-use diary Profiles of Occupational Engagement POES	Content analysis	H	The study showed that social interaction and occupations formed the two foundations of the day centres
Tsang, Ng [57]	Effects of the ‘clubhouse’ model of rehabilitation on various psychosocial issues for Chinese patients with schizophrenia living in the community	Hong Kong	Mean age = 40.5; males and females	Chronic schizophrenia	longitudinal, case–control and naturalistic design/Quantitative	Demographic and clinical variables Positive and Negative Syndrome Scale Beck Depression Inventory World Health Organization Quality of Life—Brief Version Rosenberg Self-Esteem Scale Levenson Internality, Powerful Others and Chance Scale of Locus of Control	Descriptive and inferential statistics		The program improved the psychological, social relationship and environmental quality of life of participants
Tondora, O’Connell [58]	Rationale, design, and lessons learned during the implementation of a randomized clinical trial testing the effect of using peer facilitative advocates to promote culturally responsive person-centred care planning on QOL variables, community connections, and coping for people of colour with psychotic disorders	USA	Mean age = 43.5; males and females	Schizophrenia, schizoaffective disorder, or affective disorder	Randomized clinical trial/Quantitative	Treatment Planning Questionnaire Sense of Community Index NEO-Five-Factor Inventory Multigroup Ethnic Identity Measure Scale of Ethnic Experiences Africultural Coping System Inventory Brief COPE Working Alliance Inventory—Short Form Revised (WAISFR) Health-Care Climate Questionnaire Recovery Self-Assessment Full Harmonized Social Capital Inventory Empowerment Scale Hope Scale Interpersonal Support Evaluation List Rosenberg Self-Esteem Scale MHSIP Consumer Satisfaction Survey QOL interview Paranoid Ideation and Psychoticism Subscales of the Symptom Checklist (SCL)-90 SCL-90 anxiety dimension Global Assessment of Functioning—Modified Version	Descriptive and inferential statistics	H	The project suggests the need to make cultural modifications, longer engagement period with participants
Twamley, Vella [59]	To evaluate the efficacy of supported employment for middle-aged or older people with schizophrenia	USA	Mean age = 51; males and females	Schizophrenia or schizoaffective disorder aged	Randomized controlled trial/Quantitative	UCSD Performance-Based Skills Assessment (UPSA) Positive and Negative Syndrome Scale (PANSS) Hamilton Rating Scale for Depression	Descriptive and inferential statistics	H	Individual Placement and Support (IPS) was effective for people with schizophrenia compared with conventional vocational rehabilitation (CVR)
Waghorn, Dias [60]	This investigation compared the utility of two approaches to measuring the effectiveness of a supported employment program	Australia	Mean age = 34.1; males and females	Schizophrenia, schizoaffective disorder, schizophreniform disorders, bipolar affective disorder, major depression and anxiety disorders	Non-randomized trial/Quantitative	IPS fidelity scale Socially Valued Role Classification Scale (SRCS) Scale for the Assessment of Positive Symptoms (SAPS) Scale for the Assessment of Negative Symptoms (SANS) Demographic information	Descriptive and inferential statistics	H	The non-RCT IPS cohort were more effective in gaining competitive employment compared with RCT IPS
Whitley, Harris [61]	Explore and elucidate whether components of these communities appeared to assist recovery from the point of view of consumers, and if so which were the most important factors	USA			Grounded theory approach/Qualitative	Focus groups Observations	Grounded theory approach	H	The community is perceived as a place of safety, surrogate family, socialization and individual growth
Zemore and Kaskutas [62]	explores whether services received differed by program modality (i.e., day hospital vs. residential)	USA	Age = ≥18; males and females	Alcohol-dependent only, drug-dependent only, alcohol and drug dependent	Randomized controlled trial/Quantitative	Treatment Services Review (TSR) Demographics and other covariates	Descriptive and inferential statistics	H	Residential participants showed greater participation in sober recreational events and informal socialization with peers. Higher participation in optional or extracurricular 12-step meetings was associated with better treatment outcomes
Zhou, Zhou [63]	Effectiveness of the rehabilitation services provided at the ‘Sunshine Soul Park’ on the psychotic symptoms and social functioning of individuals with schizophrenia	China	Mean age = 39.2; males and females	Schizophrenia	Non-randomized controlled trial/Quantitative	PANSS Quality of Life Inventory-74 (GQOLI-74) Social Disability Screening Schedule (SDSS) Insight and Treatment Attitude Questionnaire (ITAQ)	Descriptive and inferential statistics	H	The intervention is effective in improving the social functioning of patients with schizophrenia and in helping them understand and manage their illness

**Table 3 ijerph-18-08873-t003:** Themes identified from mixed-methods synthesis.

Global Themes	Organizing Themes	N	Papers
Context or environment for implementing recovery services	Community based	22	[24,27,29,32,37,38,40,45,46,47,48,50,51,53,54,55,57,58,59,60,61,63]
	Residential facilities	7	[27,28,29,35,36,41,62]
	Home-based care or services	4	[33,34,36,45]
	Day centre-structured program	3	[28,37,56]
	Psychiatric day hospital/primary services	6	[25,26,30,31,44,57]
Mechanisms for implementing recovery services	Integrated recovery services	17	[24,25,26,30,34,40,44,45,46,47,48,53,56,58,61,62,63]
	Vocational rehabilitation	18	[27,28,29,33,34,37,38,39,42,43,51,54,55,56,57,59,60]
	Recovery narrative photovoice and art making	5	[32,35,41,49,50]
Outcome of recovery services	Psychiatric medication	10	[24,25,33,36,42,44,48,53,62,63]
	Improving functionality	14	[24,26,36,37,39,41,43,44,45,48,51,53,55,63]
	Reduce symptoms	11	[25,30,31,36,42,44,45,46,53,57,63]
	Improving physical health and social behaviour	7	[24,26,36,45,46,48]
	Economic empowerment	19	[24,25,26,27,28,29,31,33,34,36,38,42,43,47,51,56,57,59,60]
	Household integration	3	[24,26,36]
	Social inclusion (community integration)	27	[24,25,26,30,31,32,33,34,35,37,38,39,41,43,44,45,49,51,52,54,55,56,57,58,61,62,63]
	Social support	7	[31,33,34,36,39,50,61]
	Self-efficacy	21	[24,25,26,30,32,33,34,35,38,39,40,43,47,49,50,51,52,54,55,57,61]

**Table 4 ijerph-18-08873-t004:** Mechanisms for implementing recovery services.

Recovery Services	Intervention	How to Deliver the Intervention	Process or Outcome
Integrated recovery services	Illness management	Rehabilitation training (hourly, days and weekly) meetings for adult living with severe mental illness and family members Day treatment, medication monitoring and intellectual activities Cognitive behaviour therapy techniques, psychoeducation, relapse prevention, social skills training, spiritual intervention Inclusion of spiritual intervention in conventional rehabilitation services Attending didactic and counselling sessions	Antipsychotic medication adherence, improve knowledge, decrease relapse, reducing inpatient and crisis services
	Mindfulness-based interventions	Mindfulness group therapy sessions (training hourly, daily and weekly) Meetings for consumers and family members	Improving functioning, symptoms and quality of life
	Task-sharing or task-shifting approaches	Participatory training for non-specialists to provide mental health services in communities Refer adults living with severe mental illness to primary health care and specialist services	Increase access to psychiatric medication, improve physical health and social behaviour, self-efficacy
	Home visits concept	Weekly home visits by mental health professionals to deliver mental health education, advocacy, community outreach and community orientation Adults living with severe mental illness and family members are taught about where and how to buy medication Reminding consumers and families to attend follow up	Household integration, improving self-care or practical skills
	Music-creation therapy	• A 90 min music-creation group activity organized weekly for 32 consecutive weeks	Community acceptance, inclusion, empowerment
	Active leisure or recreational activities	• Playing sports, opportunity to play games, eat, and socialize, embarking on excursions and relaxation	Social inclusion (community integration, social contacts or social interactions and socialization)
	Everyday life rehabilitation	Weekly meeting for approximately 1 to 2 h Set personal goals such as regular walks, weekly sauna bathing, social interaction with friends, initiating small talk with women, eating lunch in a restaurant, healthy cooking, and taking control of one’s own pocket money	Self-efficacy (self-esteem or self-confidence)
Vocational rehabilitation	Individual Placement and Support (IPS)	Initial vocational assessments Job searching Individual job development Training and regular supervision Work performance monitoring Support for employers and continuing post-employment support	Returning to open employment, gaining competitive employment, economic empowerment, gaining financial literacy skills, financial independence and stability
	Conventional vocational rehabilitation (e.g., sheltered employment, supported employment enterprises and social firms)		Economic empowerment
Narrative photovoice and art making	Photovoice	Taking of photographs from daily life and constructing text, exhibition and large group discussion Exploring, documenting and sharing ideas about recovery Weekly class sessions and community outings and exercises on construction of recovery	Empowerment, community integration, hope, progress in recovery
	Art making and exhibition	Weekly art appreciation class took place for several months Art appreciation class includes both classroom sessions and community outings	Development of new skills, community integration

## Data Availability

All data generated or analysed during this study are included in this article and its Appendix A files.

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
