# Peer review of "An Integrative Review of Recovery Services to Improve the Lives of Adults Living with Severe Mental Illness"

_ijerph, 2021, doi:10.3390/ijerph18168873_

Round 1

Reviewer 1 Report

This review synthesizes studies published in recent years about the personal recovery of people suffering from serious mental illnesses. The methodology is well thought out and provides data that could be especially useful for mental health professionals.

However, each included study uses different tools for assessing the clinical and personal recovery of the included patients, which is a limitation that does not allow us to distinguish which intervention is most useful. In addition, adults with different mental illnesses have been included, who probably have different prognoses in the different intervention programs.

2.3 Exclusion criteria: "The review excluded articles published prior to 2008 as well as non-English language articles": This is already defined in the inclusion criteria

Table 2. Column 5 "Participants": Mental disorder/Mental Illness maybe is a more appropriate title for this column

Table 2. Column 9: It would be interesting to reflect the level of significance of the studies outcomes

Author Response

August 9, 2021

Dear Editor,

Response to Reviewers Query regarding the manuscript: “An integrative review of recovery services promoting personal recovery

of Adults living with Severe Mental illness.

Thank you for the opportunity to submit the above manuscript for consideration in your reputable journal. Please I write on behalf of the authors to submit a response to the queries provided in the above manuscript. The authors have responded to the queries and highlighted them in yellow ink in the revised version of the manuscript. The reviewer's query has been addressed as follows:

Query

Response

Reviewer 1

However, each included study uses different tools for assessing the clinical and personal recovery of the included patients, which is a limitation that does not allow us to distinguish which intervention is most useful. In addition, adults with different mental illnesses have been included, who probably have different prognoses in the different intervention programs.

Thanks for the suggession. The authors have included these issues in the limitations of the study.

2.3 Exclusion criteria: "The review excluded articles published prior to 2008 as well as non-English language articles": This is already defined in the inclusion criteria

Thanks for the queries. The authors have removed this sentence from the exclusion criteria section

Table 2. Column 5 "Participants": Mental disorder/Mental Illness maybe is a more appropriate title for this column

The authors acknowledge the query. However, the authors feels that the title should describe the chharacteristics of included papers, rather than the participants.

Table 2. Column 9: It would be interesting to reflect the level of significance of the studies outcomes

The authors acknowledge the suggestion by the reviewer. However, since we did not perform meta-analysis, it would be difficult to estimate or quantify the level of signifance of the studies outcomes.

Thanks and looking forward to hearing from you.

Reviewer 2 Report

REVISION ID: ijerph-1263766

I found the manuscript well written and appropriate to the aim and scope of the Journal. The interest in recovery services for adults with severe mental illness is an important aspect of a patient’s independence.

The integrative format developed by the authors included different areas of recovery in a whole frame: developing independent living skills, improving quality of life, community mobilisation, reducing inpatient and crisis services, adhering to treatment and setting meaningful goals towards recovery.

The method section is completed following the different protocols published such as PRISMA, or the use of the Mixed Methods Appraisal Tool (MMAT) and the Joanna Briggs Institute (JBI), which I have found adequate.

The most important conclusion is that the recovery services are reported to be relevant in areas such as medication and treatment adherence, functioning, symptoms, physical health & social behaviour, self-efficacy, economic empowerment, social inclusion, community integration, household integration and access to social support services.

This conclusion has a great implication in social policies in relation to the recovery of adults with serious mental illnesses. It highlights the need for multidisciplinary work with this type of patients, and that the ultimate goal is their independence.

My only concern in the manuscript is just a single advice: the convenience of using the corresponding “doi” for any reference.

Author Response

August 9, 2021

Dear Editor,

Response to Reviewers Query regarding the manuscript: “An integrative review of recovery services promoting personal recovery

of Adults living with Severe Mental illness.

Thank you for the opportunity to submit the above manuscript for consideration in your reputable journal. Please I write on behalf of the authors to submit a response to the queries provided in the above manuscript. The authors have responded to all the reviewers' queries and highlighted them in yellow ink in the revised version of the manuscript. The reviewer's query has been addressed as follows:

Query

Response

Reviewer 2

My only concern in the manuscript is just a single advice: the convenience of using the corresponding “doi” for any reference.

We expect that the “doi” will be used when the paper is published by the journal.

Please could you be specific on what you mean by the convenience of using the corresponding "doi" for any reference?

Reviewer 3 Report

There are a number of issues with this article.

First of all the title is misleading personal recovery means improvement not recovery and therefore recovery should not be used if improvement is meant.   Secondly the abstract is not a reflection of the article and needs to be completely rewritten.   The article is not a review, it simply copies and pastes what a number articles have stated in their article irrespective of if those articles come to the right conclusion or not.    It's also unclear why articles from before 2008 were excluded.    The strengths and weaknesses do not reflect the strengths and weaknesses of the article.    As noted by the authors most studies in their review are about patients with schizophrenia. As everybody knows those patients do not recover from it. And as noted when talking about personal recovery, when improvement is meant, is simply misleading.   

Author Response

August 9, 2021

Dear Editor,

Response to Reviewers Query regarding the manuscript: “An integrative review of recovery services promoting personal recovery of Adults living with Severe Mental illness.

Thank you for the opportunity to submit the above manuscript for consideration in your reputable journal. Please I write on behalf of the authors to submit a response to the queries provided in the above manuscript. The authors have responded to all queries and highlighted them in yellow ink in the revised version of the manuscript. The reviewer's query has been addressed as follows:

Query

Response

Reviewer 3

First of all the title is misleading personal recovery means improvement not recovery and therefore recovery should not be used if improvement is meant. As noted by the authors most studies in their review are about patients with schizophrenia. As everybody knows those patients do not recover from it. And as noted when talking about personal recovery, when improvement is meant, is simply misleading.

Thanks for the queries. The authors wish to to clarify that personal recovery is appropriate as used in the title. The reason being that the concept of recovery could be classified into clinicial and personal. The clinical recovery considers the objective outcome, which is based on medications and treatment. Contrary, the personal recovery concept has been identified in the literature as a process, thus, there is no end. The main focus of the paper is to synthesis papers on specific interventions and approaches used to achieve this personal recovery process. This aspect of recovery has its values, principles and specific interventions. Given that these consumers may not have complete clinical recovery, there is an improvement in the personal recovery process.  The empirical papers included in this review have been piloted and implemented  using these participants group. We believe that, based on the included papers, the concept of personal recovery as used in the article is not misleading. There is an increasing papers, attempting to clarify the meaning of personal recovery as a process or an outcome.

Secondly the abstract is not a reflection of the article and needs to be completely rewritten.  

Thanks for your query. The authors wish to clarify that the abstract present findings on the main research question/objective. It starts with the objectives, methodology, and the results of the review. The first phase of the results present the characteristics of the included papers, followed by the various intervention and services identified that promote personal recovery. The final phase of the abstract presents findings on the effectiveness or outcome of these services.

The article is not a review, it simply copies and pastes what a number articles have stated in their article irrespective of if those articles come to the right conclusion or not.   

Thanks for comment. The authors wish to clarify that the article is a review. The type of review we conducted is an integrative review. The process and rationale for conducting this integrative review is presented in the methodology. We also wish to clarify that the integrative review approach is different from the different types of review (eg. traditional literature review, systematic review, scoping review, and narrative reviews).

It's also unclear why articles from before 2008 were excluded.   

Thanks for the query. We wish to clarify that we justified the reason for limiting the review to papers published from 2008. For example, in the inclusion criteria section, we indicated that the review considered only studies published in the English language. Studies published from January 2008 to January 2020, were considered for inclusion in this review. This year appears as the period where researchers’ increasingly attempted to research into recovery services and interventions for an adult with mental illness.

The strengths and weaknesses do not reflect the strengths and weaknesses of the article.   

Thanks for the comment. We wish to clarify that the strength and weaknesses of the article reflect the actual strength and weakness. We would also be grateful if you could be specific about any strength and weakness that is misrepresented.

Round 2

Reviewer 3 Report

Dear authors,    You class this review as an integrative review. However, an  "integrative literature review is a form of research that reviews, critiques, and synthesizes representative literature on a topic in an integrated way such that new frameworks and perspectives on the topic are generated." And as I said before there is no form of critique or critical analysis in this article.   Secondly, the use of the term Recovery is misleading. What they actually mean is improvement and they should state that because that's something totally different. A good example of that is when they say that "the recovery services used to promote recovery among adults living with severe mental illness." Whereas everybody knows that those adults with severe mental illness, and the articles reviewed in this review are mostly about schizophrenia, will never recover.    The sentence about excluding articles published prior to 2008 as well as non English language articles has been removed but I don't understand that. So suddenly articles in a different language and articles from prior to 2008 have been included?   Yet in limitations you say that articles from prior to 2008 are still excluded. If that is the case then you need to explain earlier in the article why studies from before 2008 were excluded. Nothing more than that. And that articles which were not in English were excluded is logical as English is the language of the article.    It's important to make it clear in the abstract and discussion and conclusion that most of the included articles are about schizophrenia.    Limitations:  there is no critical review of any of the articles included in the review;  no objective outcomes are used (to define recovery).    Some of the other limitations of the article are simply caused by the lack of quality of some of the studies in the review.    As an example, according to you, the article by Zhou, shows that the intervention is effective in improving social functioning of patients. It is unclear what the treatment actually entails, also it's a non-randomised trial. Which means that you cannot come to that sort of conclusion.    Apparently according to the review, the recovery services have helped to improve economic empowerment of the adults with severe mental illness yet you do not come up with any figures before and after treatment to backup this claim.    This review will only add something to the literature, when you have a critical look at the articles included, instead of copying and pasting what the authors of the original article have concluded. And you need to make it perfectly clear in the title, abstract and discussion / conclusion that this article is about improvement, not about recovery. You also need to make it clear that labelling improvement as recovery, as many of the articles included in the review have done, is misleading.     

Author Response

August 16, 2021

Dear Editor,

Response to Reviewers Query regarding the manuscript: “An integrative review of recovery services to improve the lives of Adults living with Severe Mental illness.

Thank you for the opportunity to submit the above manuscript for consideration in your reputable journal. Please I write on behalf of the authors to submit a response to the queries provided in the above manuscript. The authors have responded to all queries and highlighted them in yellow ink in the revised version of the manuscript. The reviewer's query has been addressed as follows:

Reviewers query

Authors response

Dear authors, You class this review as an integrative review. However, an  "integrative literature review is a form of research that reviews, critiques, and synthesizes representative literature on a topic in an integrated way such that new frameworks and perspectives on the topic are generated."

Thanks for the query. The authors wish to clarify that we have define integrative review and explain the process in the methodology. Whilst we acknowledge that integrative review can include methodological and conceptual papers, it can also be limited to empirical papers to give understanding of the topic (Whittemore, R. and K. Knaf 2005; Hopia et. al 2016; Broome 1993). We specifically followed the methodology by Whittemore et al. (2005) who point out that the review could use a narrative synthesis for qualitative and quantitative studies. The synthesis may be in the form of a table, diagram to portray results. We wish to clarify that we did not perform meta-analysis, where we could look at homogeneity or heterogeneity of included paper to determine effect size or point estimate. This is where we could make objective measure of the actual effectiveness. This is a limitation of an integrative review.

And as I said before there is no form of critique or critical analysis in this article.  

Thanks for the comment. We wish to clarify that we perform methodological quality assessment of the included papers, using the mixed method appraisal tool. For example, out of the 40 papers, 38 met the criteria for high methodological quality assessment, whilst only two papers had medium quality. Moreover, the synthesis was analytical, making comparison of individual papers, and not just descriptive. Finally, the key issues emerging from the review have been discussed in the discussion section, and further compared with relevant international literature.

Secondly, the use of the term Recovery is misleading. What they actually mean is improvement and they should state that because that's something totally different. A good example of that is when they say that "the recovery services used to promote recovery among adults living with severe mental illness." Whereas everybody knows that those adults with severe mental illness, and the articles reviewed in this review are mostly about schizophrenia, will never recover.

Thanks for the comment. We really appreciate your position on this issue. But we wish to clarify that the new concept of recovery consider recovery as a process where the consumer lives a meaningful and fulfilling life despite limitations caused by mental illness. This contradicts the traditional understanding of recovery, which perceives a complete cure. Whilst we acknowledge that consumers with schizophrenia are not going to be cured, the recovery concept seeks to give them a meaningful and fulfilling life despite the presence of illness. Consumers with longitudinal severe mental illness require a sense of hope that their recovery toward reintegrating into society is meaningful for them. Recovery is about hope more than about cure. The improvement in the condition is an attribute that is used to refer or define recovery (van Weeghel et. al, 2019). This notwithstanding, we have revised the title as “An integrative review of recovery services to improve the lives of Adults living with Severe Mental illness”

The sentence about excluding articles published prior to 2008 as well as non English language articles has been removed but I don't understand that. So suddenly articles in a different language and articles from prior to 2008 have been included?   Yet in limitations you say that articles from prior to 2008 are still excluded. If that is the case then you need to explain earlier in the article why studies from before 2008 were excluded. Nothing more than that. And that articles which were not in English were excluded is logical as English is the language of the article.   

Thanks for the query. We removed the sentence “The review excluded articles published prior to 2008 as well as non-English language articles" from the exclusion criteria section because it appears as repetition as we have already indicated in the inclusion criteria that "The review considered only studies published in the English language. Studies published from January 2008 to January 2020, were considered for inclusion in this review." This was a recommendation made by reviewer 2 during the first round of the review. Moreover, we wish to clarify that the study was limited to this period because this period appears as the period where researchers increasingly attempted to research into recovery services for an adult with mental illness.

It's important to make it clear in the abstract and discussion and conclusion that most of the included articles are about schizophrenia.   

Thanks for the comment. We have indicated this in the abstract.

Limitations:  there is no critical review of any of the articles included in the review;  no objective outcomes are used (to define recovery).    Some of the other limitations of the article are simply caused by the lack of quality of some of the studies in the review.    As an example, according to you, the article by Zhou, shows that the intervention is effective in improving social functioning of patients. It is unclear what the treatment actually entails, also it's a non-randomised trial. Which means that you cannot come to that sort of conclusion. 

Thanks for the query. We wish to clarify that we perform methodological quality assessment of the included papers, using the mixed method appraisal tool. Also, we wish to clarify that we did not perform meta-analysis, where we could look at homogeneity or heterogeneity of included paper to determine effect size or point estimate. This is where we could make objective measure of the actual effectiveness. This is a limitation of an integrative review as compared to systematic review. We have therefore acknowledged this in the limitations.

Apparently according to the review, the recovery services have helped to improve economic empowerment of the adults with severe mental illness yet you do not come up with any figures before and after treatment to backup this claim.    This review will only add something to the literature, when you have a critical look at the articles included, instead of copying and pasting what the authors of the original article have concluded.

Thanks for the comment. We wish to clarify that the findings reported in the review is a narrative synthesis of the included papers. As indicated earlier, we did not perform a meta-analysis, where we could estimate the effective size of the improvement of the outcome. This notwithstanding, we supported most of the results, with examples from individual papers. For example, regarding findings on economic empowerment, the results section presents that Most of the studies suggested that vocational intervention such as IPS is more effective than the conventional vocational services, particularly in every vocational outcome. In particular, IPS clients are more effective to work competitively, returning to open employment (eg. working for at least one day), and longer duration of employment (eg. working for many hours and longer job tenure) and wages earned (Burns et al., 2008; Catty et al., 2008; Chiu et al., 2010; Kilian et al., 2012; Koletsi et al., 2009; Twamley et al., 2012). Catty et al. (2008) reported that IPS clients were two times (214 days) more likely to work for a longer duration than vocational service clients (108 days). Conversely, 57% of IPS clients (a sample of 58 consumers with schizophrenia) worked competitively, compared with 29% of conventional vocational clients. Similarly, 70% of IPS participants obtained any paid work, compared with 36% of conventional vocational clients (Twamley et al., 2012). More importantly, the vocational rehabilitation interventions have helped consumers to gain financial literacy skills (eg. managing finances) (Asher et al., 2018; Brooke-Sumner et al., 2017; Brooke-Sumner et al., 2018; Crain et al., 2009; Panczak & Pietkiewicz, 2016), become financial independence and financially stable (Koletsi et al., 2009) and improved recovery (Asher et al., 2018; Chiu et al., 2010).

And you need to make it perfectly clear in the title, abstract and discussion / conclusion that this article is about improvement, not about recovery. You also need to make it clear that labelling improvement as recovery, as many of the articles included in the review have done, is misleading.   

Thanks for the query. We have revised the sections accordingly.
